# Toxicity Profiles of Antibody–Drug Conjugates: Synthesis and Graphical Insights to Optimize Patient-Centered Treatment Strategies for HER2-Negative Metastatic Breast Cancer

**DOI:** 10.3390/cancers17142307

**Published:** 2025-07-11

**Authors:** Bérénice Collineau, Anthony Gonçalves, Marie Domon, Damien Bruyat, François Bertucci, Alexandre de Nonneville

**Affiliations:** 1Institut Paoli-Calmettes, Department of Medical Oncology, 13009 Marseille, Francebertuccif@ipc.unicancer.fr (F.B.); 2Institut Paoli-Calmettes, CRCM (Centre de Recherche en Cancérologie de Marseille), Aix-Marseille University, 13009 Marseille, France; 3Institut Paoli-Calmettes, INSERM (Institut National de la Santé et de la Recherche Médicale), Aix-Marseille University, 13009 Marseille, France; 4Institut Paoli-Calmettes, Department of Palliative and Supportive Care, 13009 Marseille, France

**Keywords:** antibody drug conjugate, breast cancer, safety, toxicity, shared decision-making

## Abstract

The advent of antibody–drug conjugates (ADCs) has reshaped the treatment landscape for HER2-negative metastatic breast cancer through their targeted cytotoxic delivery. Trastuzumab deruxtecan (T-DXd) and Sacituzumab govitecan (SG) have shown their efficacy but come with distinct toxicity profiles, including conventional adverse events, such as hematologic and digestive toxicity, and ADC-specific adverse events like pneumonitis for T-DXd. With a growing pipeline of ADCs and limited head-to-head or sequencing data, treatment choices may become increasingly complex. In clinical scenarios where both T-DXd and SG are treatment options, understanding and comparing their safety profiles is key to guiding shared decision-making. Decision-support tools synthesizing efficacy and toxicity data are essential to help oncologists and patients select the most appropriate ADC for each clinical situation.

## 1. Introduction

Breast cancer remains the most commonly diagnosed malignancy and the leading cause of cancer-related mortality among women worldwide [1]. This disease incorporate a heterogeneous group of tumors with distinct molecular and histological subtypes, defined by the expression of hormone receptors (HRs) (estrogen receptor (ER) and progesterone receptor (PR)) and human epidermal growth factor receptor 2 (HER2) [2]. The breast cancer subtype HR-positive/HER2-negative represents the most common subtype, accounting for 70% of diagnoses, and has the best outcome compared to other subtypes [2]. For patients with HR-positive/HER-negative metastatic breast cancer (mBC), the standard first-line treatment involves endocrine therapy associated with a cyclin-dependent kinase 4/6 [3]. However, over time, most patients develop resistance to endocrine therapy, marking a pivotal turning point in metastatic disease, that necessitates the use of systemic chemotherapy; wich has variable efficacy, is often toxic and lacks specificity [3]. Triple-negative breast cancer (TNBC)—defined by the absence of ER, PR and HER2—is an aggressive subtype associated with poor prognosis that accounts for 11% of breast cancers [2]. Lacking defined molecular targets, the therapeutic landscape of TNBC is historically limited to chemotherapy, more recently with the addition of immune checkpoint inhibitors in a subset of patients with PDL1-positive tumors [3,4]. This therapeutic gap highlights the need for the development of new agents with more specificity.

Antibody–drug conjugates (ADCs) have emerged as a promising solution to these challenges and represent an innovative therapeutic class designed to deliver cytotoxic chemotherapy directly to tumor cells by targeting surface antigens via monoclonal antibodies [5]. Structurally, ADCs comprise three components: a cytotoxic payload capable of inducing apoptosis; a monoclonal antibody specific to a tumor-associated antigen; and a linker that connects the two and determines the stability and release of the cytotoxic agent within the tumor microenvironment. ADCs aim to enhance tumor targeting, minimize off-target toxicity and thereby improve the therapeutic index. One important pharmacologic parameter is the drug-to-antibody ratio (DAR), which varies between ADCs and influences potency and toxicity. The efficacy of an ADC appears to depend on factors specific to the antibody, the linker, and the payload, each of which is subject to complex interactions between the ADC and various components of the tumor and its microenvironment [6].

Although the concept of ADC was introduced in the 1980s, early-generation ADCs failed to demonstrate significant clinical efficacy due to suboptimal linker chemistry, poor internalization, and off-target toxicity [7,8]. It was not until 2013 that ADCs reappeared in the solid tumor oncology therapeutic field with ado-trastuzumab emtansine (T-DM1), becoming the first ADC targeting HER2 to be approved by the FDA in HER2-positive mBC [9]. This paved the way for further innovations and a new generation of ADCs with improved linkers, more potent payloads, and better intracellular trafficking profiles.

More recently, two ADCs have significantly altered the treatment landscape of HER2-negative breast cancer: trastuzumab deruxtecan (T-DXd) and Sacituzumab govitecan (SG). T-DXd initially demonstrated its efficacy compared to standard-of-care cytotoxic chemotherapy in HER2-positive mBC [10,11,12] and is now approved for HR-positive/HER2-negative mBC and mTNBC based on the results of the phase 3 DESTINY BREAST 04 (DB04) and DESTINY BREAST 06 (DB06) trials (1) [13,14]. The DB04 trial included patients (mostly with HR-positive but also with HR-negative tumors, in a small subset of them) with “HER2-low” mBC, defined as HER2 expression with a score on immunohistochemical (IHC) analysis of 1+, or a score of 2+ and negative results on in situ hybridization (ISH). In contrast, the DB06 trial expanded the potential eligibility to include patients with “HER2-low” and “HER2-ultralow” mBC, the latter defined as HER2 IHC 0 with faint membrane staining. SG, on the other hand, first demonstrated its efficacy compared to standard-of-care cytotoxic chemotherapy in mTNBC through the phase 3 trial ASCENT and later in HR-negative/HER2-positive mBC with the phase 3 TROPIC-02 trial [6,15,16].

Despite these advances, ADCs are not devoid of toxicity [17]. While designed to offer improved selectivity, they still exhibit adverse event (AE) profiles comparable to conventional chemotherapy, and in some cases introduce unique toxicities, such as interstitial lung disease (ILD), seen with T-DXd, or severe neutropenia and diarrhea with SG. Consequently, the implementation of ADCs in clinical practice requires a nuanced understanding of their efficacy–toxicity balance.

In an era of personalized medicine, where treatment decisions increasingly incorporate patient values and preferences, the concept of shared decision-making (SDM) becomes particularly relevant. The National Academy of Medicine defines patient-centered care as “care that is respectful of and responsive to individual patient preferences, needs, and values […], that patients values guide all clinical decisions”. In situations where the expected benefits are uncertain and quality of life outcomes differ across treatment options, patients should be involved actively in the decision [18]. Importantly, DestinyBreast 04 and TROPICS-02 have included patient-reported outcomes (PROs), which offer valuable insight into the impact of these two ADC on symptoms, global health status, and functional well-being [14,15]. These data support individualized treatment choices aligned with patients’ values and expectations. SDM has already shown some evidence regarding the improvement of affective–cognitive outcomes [19]. In the PROSPERO study, describing treatment options was the most relevant component in different SDM models [20]. Value assessment tools for patients with mBC are being developed to aid in treatment decisions and communication [21]. Moreover, decision aids incorporating visual elements are being developed in the context of SDM, especially for underserved patient populations [22].

Given the growing availability of ADCs for HER2-negative mBC, oncologists are increasingly faced with decisions between T-DXd and SG, particularly in late-line settings where both may be viable options. A systematic, comparative understanding of their respective toxicity profiles is therefore essential to inform both clinicians and patients. Furthermore, considering the psychosocial and physical burdens of advanced breast cancer, empowering patients to engage meaningfully in therapeutic choices may improve adherence and their quality of life. In this study, we aimed to synthesize and graphically represent the toxicity profiles of T-DXd and SG based on data from pivotal phase 3 trials, focusing on HR+/HER2-negative and triple-negative populations. Our goal is to provide clinicians with accessible tools to support SDM and help patients navigate complex therapeutic landscapes with greater confidence and clarity.

## 2. Material and Methods

We chose to evaluate ADC toxicity from prospective phase 3 trials in HR-positive/HER2-negative mBC and mTNBC. The toxicity profiles, including frequency and severity, for T-DXd and ASCENT/TROPICS-02 for SG were extracted from DB04/DB06 and analyzed. Descriptive analyses were performed; this included reports of the percentage of each adverse event (AE) of interest and weighted means (wMean) for each AE, minimizing bias related to differences in the sample sizes of the clinical trials. We used a descriptive pooled analysis of adverse event frequencies, aggregating raw event counts and patient numbers across studies without formal hypothesis testing [23]. Weighted means were calculated as follows: wAE = (nAEs1 + nAEs2)/(Ns1 + Ns2), with wAE being the weighted mean of the AE of interest, nAE the number of AE in interest in study 1 (s1) and study 2 (s2), and Ns1 and Ns2 being the number of evaluable patients in study 1 and study 2, respectively. The visualization of radar plots was generated in Python using the matplotlib and numpy packages (Python 3.9 à 3.12 et Matplotlib 3.6).

## 3. Trastuzumab-Deruxtecan (T-DXd)

T-DXd is an ADC consisting of the anti-HER2 monoclonal antibody trastuzumab and a cleavable tetrapeptide linker. The payload is a topoisomerase I inhibitor, a derivative of exatecan, with a DAR of 8. T-DXd is administered at a dose of 5.4 mg/kg via intravenous infusion every three weeks [24,25,26].

Table 1 reports the incidence of AEs caused by T-Dxd. The most common all-grade AEs were nausea (69.2%), fatigue (47.2%), alopecia (41.9%), neutropenia (35.6%), anemia (30.5%), vomiting (30.5%), AST/ALT increase (26.7%) and anorexia (25.9%). In total, 52.7% of patients experienced grade ≥ 3 AEs. Neutropenia was the most common grade ≥ 3 AE reported (17.5%). Other grade ≥ 3 AEs were anemia (6.9%), leukopenia (6.7%) and thrombocytopenia (5.1%). LVEF alteration was rare (9.9%) and grade ≥ 3 in 1.1%.

The occurrence of pneumonitis or interstitial lung disease (ILD) induced by T-DXd has been described in phase 1/2 studies, with some rare fatal AEs [27,28,29]. We report PID in 10.7% of patients, with 2.6% experiencing grade ≥ 3 PID. Importantly, only 0.8% presented grade 5 PID in DB04 and 0.7% presented grade 5 PID in DB06 [13,14]. The pathophysiology of T-DXd-induced ILD remains unknown. The median time to ILD onset was 4.3 months [range: 0.9; 23.7] in Destiny Breast 03 [12]. Although there is no ILD-specific absolute counterindications to initiate T-DXd, it is important to evaluate patient characteristics, especially the risk factors for developing ILD that have been described. These include Japanese population, dose, baseline peripheral oxygen saturation, moderate or severe renal impairment, lung comorbidities, obesity and time since initial diagnosis [30].

### 3.1. Management of ILD

Although ILD is most often of low grade (grade ≤ 2), precautions with multidisciplinary involvement are required in clinical practice and should include the consideration of referral to a pneumologist. Recommendations for the management of T-DXd-related ILD were actualized in 2023 [30]. Systematic screening via computed tomography (CT) of the chest without contrast is recommended before starting T-DXd and should be repeated in the first year least every 12 weeks and every 6–9 weeks in the case of respiratory baseline symptoms. In case of suspected PID, a high-resolution CT chest is recommended. A pulmonary functions test can be considered, but there is no evidence that his has a predictive impact [30].

In cases of asymptomatic ILD (grade 1), it is recommended that T-DXd therapy is interrupted and that oral corticosteroid treatment at 0.5 mg/kg is considered. T-DXd can be reintroduced at the same dose if it is resolved within 28 days. T-DXd can be reintroduced at a reduced dose if resolved in >28 days: 4.4 mg/kg can be used as the first reduction, and 3.2 mg/kg can be used as the second dose reduction. T-DXd must be discontinued if ILD is not resolved within 49 days of the last infusion. Close monitoring should be performed, with a clinical assessment within 2–7 days and follow-up imaging at 7–14 days [30].

In the case of symptomatic ILD (grade 2), T-DXd should be permanently discontinued; after this, corticosteroid oral treatment at ≥1 mg/kg for at least 14 days or until the complete resolution of clinical and chest CT findings is recommended, followed by gradual taper over ≥4 weeks [30].

In the case of worsening, with grade 2 or grade 3 ILD or greater, hospitalization is required to promptly initiate high-dose intra-veinous corticosteroid treatment: 500–1000 mg for 3 days followed by ≥1 mg/kg for at least 14 days, or until complete resolution with gradual taper over >4 weeks. If symptoms are not improved within 3–5 days, immunosuppressive therapy (e.g., infliximab, mycophenolate mofetil) or local treatment may be considered after the elimination of alternative etiologies. [30]

### 3.2. Prophylaxis of AE for T-DXd

The primary or secondary prevention of these adverse events is essential for managing T-DXd. Although T-DXd was not classified as highly emetogenic in pivotal trials, in clinical practice, the use of antiemetic regimens appropriate for highly emetogenic agents—such as 5HT3 receptor antagonists (setrons), NK1 receptor antagonists (e.g., aprepitant), and corticosteroids [31]. Since neutropenia is often of a low grade, the routine use of G-CSF is not recommended [31]. The risk of alopecia may be reduced with the use of a cooling cap [31].

## 4. Sacituzumab Govitecan (SG)

Sacituzumab govitecan (SG), also known as IMMU-132, is composed of a humanized anti-TROP2 (trophoblast cell-surface antigen 2) antibody linked via a cleavable linker to the cytotoxic payload SN-38, the active metabolite of irinotecan, a topoisomerase I inhibitor with a DAR of 7.5 to 8 [32]. Trop-2 is a transmembrane glycoprotein receptor encoded by the TACSTD2 gene, which allows for calcium signal transduction intracellularly, playing a role in tumor progression [6]. SG is metabolized by the UGT1A enzyme (uridine diphosphate glucuronosyltransferase 1A1), whose interindividual variability, particularly among different ethnic groups, can lead to increased toxicity, notably in individuals with homozygosity for UGT1A1*28, which is associated with reduced enzyme activity [32]. However, routine genotyping is not yet practiced. The half-life of SG is approximately 10.3 to 11.4 h, and it is eliminated via the renal route. SG is administered at a dose of 10 mg/kg via intravenous infusion on days 1 and 8 of each 21-day cycle [25,26].

Table 2 reports the incidence of AEs associated with SG. The most common all-grade AEs were neutropenia (67.1%), diarrhea (60.8%), nausea (58.7%), alopecia (46.4%), fatigue (44.5%), anemia (36.7%), constipation (27.8%) and vomiting (26%). Any adverse event of grade ≥ 3 occurred in 69% of cases. Neutropenia was the most common grade ≥ 3 AE reported (51.3%). Diarrhea (10.3%), leukopenia (9.3%), anemia (7.8%) and febrile neutropenia (5.5%) were other common grade ≥ 3 AEs.

### 4.1. Prophylaxis of AE for SG

Given this toxicity profile, recommendations have been developed for managing side effects under SG [33]. The cholinergic syndrome induced by govitecan can be prevented by administering atropine before infusion. Optimizing patient education regarding the management of diarrhea is necessary before starting treatment and should be reiterated if symptoms appear. It is recommended that antiemetic prophylaxis is used before injection with 5HT3 antagonists and dexamethasone. NK1 antagonists can be considered in patients at risk of chemotherapy-induced nausea and vomiting. If necessary, ondansetron or prochlorperazine can be prescribed in the days following treatment. For anticipatory or refractory nausea, olanzapine may be an option. Hypersensitivity reactions are prevented by premedication with antipyretics and H1 or H2 antihistamines before injections.

### 4.2. Management of AE

In the event of severe neutropenia—defined as grade 3 febrile neutropenia, grade ≥ 4 neutropenia lasting less than one week, or grade ≥ 3 neutropenia resulting in treatment delay—it is recommended that the dose of SG is reduced by 25% and that secondary prophylaxis with G-CSF is initiated [33]. G-CSF can be administered for 2 to 3 consecutive days between Day 4 and Day 6 if neutropenia occurs on Day 1 of the treatment cycle, or as a single dose in pegylated form on Day 9 or Day 10 if neutropenia occurs on Day 8 [33]. If there is a relapse of severe neutropenia, a secondary dose reduction of 50% is recommended. Treatment should be discontinued if there is a second relapse [33]. Primary prophylaxis for neutropenia with G-CSF is not routinely recommended.

### 4.3. Graphical Representations Propositions to Guide Therapeutic Decision-Making

The optimal treatment strategy for ADCs in HR-positive/HER2-negative breast cancer and TNBC in the metastatic setting remains undefined [34,35]. Although T-Dxd and SG share a similar payload, their distinct molecular features result in both overlapping and specific AEs. In the context of SMD-making, selecting between two potentially effective treatments could be driven by the toxicity profiles patients are willing to accept and manage. This decision considers personal, psychological, and societal factors, as well as comorbidities.

Figure 1 and Figure 2 provide a non-comparative representation of the incidence rates of AEs between these two ADCs in DB04/06 and ASCENT/TROPICS-02. Radar plots are designed for easy patient interpretation, providing a clear overview of the toxicities associated with each treatment. This tool could empower patients to actively contribute to the medical decision-making process based on their ability to handle potential AEs.

## 5. Discussion

ADCs have emerged as a transformative addition to the therapeutic landscape for HER2-negative breast cancer, particularly in metastatic settings. However, the clinical application of ADCs such as T-DXd and SG highlights significant challenges.

Firstly, the toxicities associated with ADCs, including neutropenia, gastrointestinal symptoms, and more specifically ILD, require careful monitoring and thorough management. Here, we reported data from clinical trials, but additional real-world data on toxicities and their management in clinical practice would be valuable. Despite the high prevalence of any-grade AE (99% with T-DXd and 98% with SG), both agents were associated with a reduced risk of definitive deterioration compared to chemotherapy [14,15]. While toxicity profiles and SDM are critical considerations, it is also essential to recognize the heterogeneity of patient populations across the pivotal trials analyzed in our study (HR+/HER2-negative and TNBC).

Secondly, the increasing number of ADCs that have notable toxicity profiles, such as Datopotamab Deruxtecan (Dato-DXd), targeting TROP2, and Patritumab Deruxtecan (HER3-DXd), targeting HER3, introduces complexity in determining the optimal sequencing, potential combinations, and selection of ADCs, particularly in patients eligible for multiple agents [34,36].

While both ADCs mentioned demonstrated promising efficacy, they present notable toxicity profiles; this includes ILD, with Dato-DXd also being associated with specific AEs such as stomatitis and keratitis, which may further increase the complexity of decision-making regarding treatment [37,38,39,40]. As ADCs are increasingly integrated into the treatment landscape across all breast cancer subtypes, including in early-stage and potentially curative settings, it becomes essential to thoroughly characterize AEs of special interest—such as ILD—when these agents are administered earlier in the disease course. Anticipating and managing such toxicities in patients with fewer comorbidities and potentially longer life expectancies presents additional challenges for oncologists.

SMD can play a crucial role in selecting treatment options in oncology, particularly given the complexity and impact of these choices on quality of life [41]. Using visual tools to present adverse event profiles could empower patients to weigh the benefits and risks according to their personal priorities and tolerance thresholds, fostering a truly individualized approach to care. This approach is also encouraged by the French administration (HAS) [42] but requires a prospective assessment regarding utility and acceptability. It is crucial to gather real-world data, as findings from clinical trials may only partially represent what is observed in routine practice. Beyond clinical trial data, treatment decisions must integrate the heterogeneity of the patient populations studied, as well as individual patient characteristics, treatment goals, and personal values. Clinicians must comprehensively manage the patient by addressing all symptoms and factors related to quality of life (psychological, professional, social, intimate), actively inquiring about any side effects she may hesitate to disclose, such as vaginal dryness, decreased libido, or localized pain, which can impact her intimate and feminine well-being.

## 6. Conclusions

Ultimately, the evolving field of ADC holds promise regarding the significant improvement of outcomes in metastatic breast cancer, offering new therapeutic options beyond traditional chemotherapy. As these agents expand to earlier lines of treatment and to broader patient populations, including those with HER2-low and triple-negative subtypes, optimizing patient selection and management will become increasingly critical. Future research should prioritize the development of personalized approaches to ADC therapy, incorporating patient comorbidities, treatment history, quality of life considerations and individualized tolerability thresholds. A deeper understanding of biomarkers predicting both efficacy and toxicity will be key to refining therapeutic algorithms. Finally, integrating patient preferences into SDM processes will be essential to ensure that treatment choices align with each patient’s values and life goals, particularly in this complex and evolving therapeutic landscape.

## Figures and Tables

**Figure 1 cancers-17-02307-f001:**
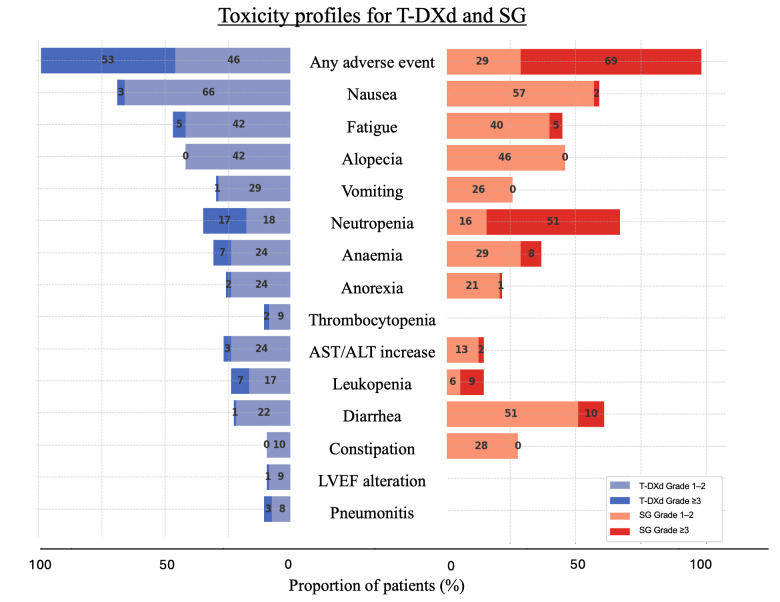
Diverging bar chart—non-comparative representation of toxicity profiles (selected AE incidences) for T-DXd and SG in pivotal trials.

**Figure 2 cancers-17-02307-f002:**
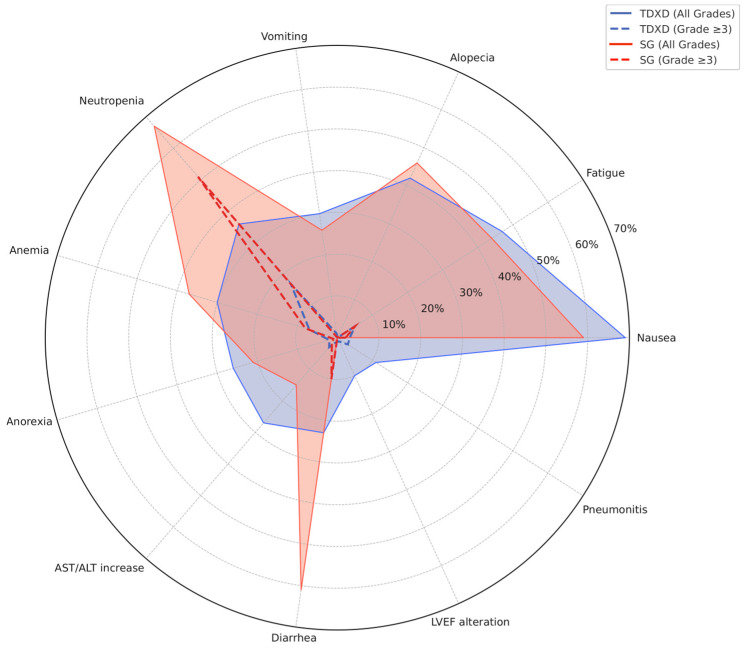
Radar plot—non-comparative representation of toxicity profiles (selected AE incidences) for T-DXd and SG in pivotal trials.

**Table 1 cancers-17-02307-t001:** Safety profile of T-Dxd (all-grade AEs and grade ≥ 3 in proportion (%)).

	All Grade (%)			Grade ≥ 3 (%)		
	wMean	DB04	DB06	wMean	DB04	DB06
Any adverse event	99.1	99.5	98.8	52.7	52.6	52.8
Nausea	69.2	73	65.9	3.0	4.6	1.6
Fatigue	47.2	47.7	46.8	5.5	7.5	3.7
Alopecia	41.9	37.7	45.4	0	0	0
Neutropenia	35.6	33.2	37.6	17.5	13.7	20.7
Anemia	30.5	33.2	28.1	6.9	8.1	5.8
Vomiting	30.3	34	27.2	1.4	1.3	1.4
AST/ALT increase	26.7	23.5	29.5	2.7	3.2	2.3
Anorexia	25.9	28.6	23.5	1.9	2.4	1.4
Thrombocytopenia	23.7	23.7	NA	5.1	5.1	NA
Leukopenia	23.3	23.2	23.3	6.7	6.5	6.9
Diarrhea	23.1	22.4	23.7	1.5	1.1	1.8
Constipation	21.3	21.3	NA	0.0	0	NA
Pneumonitis	10.7	10	11.3	2.6	4.1	1.4
LVEF alteration	9.9	11.9	8.1	1.1	1.5	0.7

Calculated wMean and % described in each study.

**Table 2 cancers-17-02307-t002:** Safety profile of SG (all-grade AEs and grade ≥ 3 in proportion (%)).

	All Grade (%)			Grade ≥ 3 (%)		
	wMean	ASCENT	TROPICS-02	wMean	ASCENT	TROPIC-02
Neutropenia	67.1	64	70	51.3	52	51
Diarrhea	60.8	65	57	10.3	11	9
Nausea	58.7	62	55	2.1	3	1
Alopecia	46.4	47	46	0	0	0
Fatigue	44.5	52	38	4.9	4	6
Anemia	36.7	40	34	7.8	9	6
Constipation	27.8	37	19	0.2	0	0
Vomiting	26.0	33	19	1.0	3	0
Anorexia	21.5	28	16	1.0	2	0
Asthenia	17.7	16	20	1.7	2	2
Abdominal pain	16.9	21	13	1.7	3	1
Leukopenia	15.2	17	14	9.3	10	9
Lymphopenia	12.0	NA	12	4.0	NA	4
Neuropathy	9.0	NA	9	1.0	NA	1
Febrile neutropenia	5.5	6	5	5.5	6	5

Calculated wMean and % described in each study.

## Data Availability

The datasets used and/or analyzed during the current study available from the corresponding author on reasonable request.

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
