# Peer review of "Toxicity Profiles of Antibody–Drug Conjugates: Synthesis and Graphical Insights to Optimize Patient-Centered Treatment Strategies for HER2-Negative Metastatic Breast Cancer"

_cancers, 2025, doi:10.3390/cancers17142307_

Round 1
Reviewer 1 Report
Comments and Suggestions for Authors
The authors provide a comprehensive summary of antibody drug conjugate therapies in the treatment of HER2-negative metastatic breast cancer which is a timely topic. Some edits / comments that may improve the manuscript
*Abstract
- Background: consider re-writing this sentence: "HER2-negative metastatic breast cancer (mBC), treatment options include targeted therapies, cytotoxic chemotherapies, and immunotherapy", potentially to: The treatment options for HER2-negative metastatic breast cancer include targeted therapies, cytotoxic chemotherapies, and immunotherapy
*Introduction
- In the discussion of trials leading to T-DxD (DB04/DB06) and SG (Ascent/Tropics02), it may be helpful to note that the approvals in these trials were based on efficacy compared to a standard of care cytotoxic chemotherapy comparator
- It is important when discussing shared decision making, to emphasize we have to take into account patient reported outcomes when making therapeutic decisions, and PROs are available for these two agents from the pivotal approval trials
*Methods
- Understanding it may be challenging to do given heterogeneity of treatments in the control arms, did the authors consider doing a similar analysis for the chemotherapy arms for the phase 3 trials included in this review
- Are there references for using this methodological approach (weighted means) across trials for AE assessment?
*T-DXD
- It would be helpful to report the pneumonitis / ILD rates in the paragraph after Table 1 with the discussion on OLD. Interesting to note the higher grade 3 rates in DB04 compared to DB06, and whether differences in population / better knowledge of the drug led to that
- For the management of ILD, a couple of points
- If suspected, a high resolution CT chest is recommended
- Should include considering referral to specialist (pulmonary)
- Would include grade 5 AE numbers, as important to report
- For Nausea as an AE
- It is important to note that T-DxD is highly emetogenic, but the trials did not include management as a highly emetogenic drug. In practice, agreed as the authors note to use regimens for high emetogenicity drugs
*SG
- Please review Table 2 rows and values. It is showing grade 3 diarrhea at 51.3%, however that is the neutropenia rate. The rows may have shifted, please review
- For AE management, consider discussing diarrhea prophylaxis and management first as one of the most common side effects (more than hypersensitivity)
- For neutropenia, the last sentence prior to management of AE section should likely move down
*Discussion / Conclusions
- It may be helpful to include that while toxicity and SDM are key, it is also important to understand context and patient population. The SG trials included only TNBC patients, while the Destiny Breast trials had both HR+ and HER2- patients. That does impact decision making for clinicians, especially with the ASCENT 04 recent data of SG + pembrolizumab
- Patient reported outcomes should be discussed here, as that data is available for the Destiny Breast and ASCENT trials
- Consider noting that these ADCs are moving up in the treatment for all subtypes of breast cancer, including the curative setting. This makes it even more important to understand adverse events of special interest (like ILD) when using drugs early in the disease course
General comment: The authors are looking to implement novel AE visualization tools to facilitate SDM, and this message comes through in the final part of the paper. It would be helpful to include that message in the introduction (including prior work using this methodology), and adding some more information about this methodology in the materials and methods section
Author Response
*Abstract
- Background: consider re-writing this sentence: "HER2-negative metastatic breast cancer (mBC), treatment options include targeted therapies, cytotoxic chemotherapies, and immunotherapy", potentially to: The treatment options for HER2-negative metastatic breast cancer include targeted therapies, cytotoxic chemotherapies, and immunotherapy
We changed line 22-23 the sentence “HER2-negative metastatic breast cancer (mBC), treatment options include targeted therapies, cytotoxic chemotherapies, and immunotherapy” to “The treatment options for HER2-negative metastatic breast cancer include targeted therapies, cytotoxic chemotherapies, and immunotherapy. “
*Introduction
- In the discussion of trials leading to T-DxD (DB04/DB06) and SG (Ascent/Tropics02), it may be helpful to note that the approvals in these trials were based on efficacy compared to a standard of care cytotoxic chemotherapy comparator
We added line 89-90: “compared to standard of care cytotoxic chemotherapy”
We added lines 98-99: “compared to standard of care cytotoxic chemotherapy”
- It is important when discussing shared decision making, to emphasize we have to take into account patient reported outcomes when making therapeutic decisions, and PROs are available for these two agents from the pivotal approval trials
We added lines 114-117: “Importantly, in DestinyBreast 04 and TROPICS-02 have included patient-reported outcomes (PROs), which offer valuable insight into the impact of these two ADC on symptoms, global health status, and functional well-being [14,16].
*Methods
- Understanding it may be challenging to do given heterogeneity of treatments in the control arms, did the authors consider doing a similar analysis for the chemotherapy arms for the phase 3 trials included in this review
This point is of great interest, unfortunately, in the study included details on toxicities/AE for standard regimens are not available overall at the molecule levels but only for pooled TPC.
- Are there references for using this methodological approach (weighted means) across trials for AE assessment?
Weighted means approach is frequently employed, as illustrated by Sargent et al., who used a similar method to summarize toxicity data across trials without conducting a formal meta-analysis (J Clin Oncol 2009). We added a sentence with citation, line 141-143: “We used a descriptive pooled analysis of adverse event frequencies, aggregating raw event counts and patient numbers across studies without formal hypothesis testing”
T-DXD
- It would be helpful to report the pneumonitis / ILD rates in the paragraph after Table 1 with the discussion on OLD. Interesting to note the higher grade 3 rates in DB04 compared to DB06, and whether differences in population / better knowledge of the drug led to that
We removed the paragraph line 159 and added it lines 165-166 “We report PID in 10.7% of patients with 2.6% of grade ≥3.”
- For the management of ILD, a couple of points
- If suspected, a high resolution CT chest is recommended
We add line 181: “In case of suspected PID, a high-resolution CT chest is recommended.”
- Should include considering referral to specialist (pulmonary)
We add line 176-177 ‘’and should include considering referral to pneumologist.”
- Would include grade 5 AE numbers, as important to report
We add line 166-167 “Importantly, only 0.8% in DB04 and 0.7% in DB06 presented grade 5 PID [12,28].”
- For Nausea as an AE
- It is important to note that T-DxD is highly emetogenic, but the trials did not include management as a highly emetogenic drug. In practice, agreed as the authors note to use regimens for high emetogenicity drugs
We clarified lines 202-205:” Although T-DXd was not classified as highly emetogenic in pivotal trials, in clinical practice, the use of antiemetic regimens appropriate for highly emetogenic agents—such as 5HT3 receptor antagonists (setrons), NK1 receptor antagonists (e.g., aprepitant), and corticosteroids—is recommended.
*SG
- Please review Table 2 rows and values. It is showing grade 3 diarrhea at 51.3%, however that is the neutropenia rate. The rows may have shifted, please review
Thank you for noticing the rows were indeed shifted. We have done the modifications in the table 2.
- For AE management, consider discussing diarrhea prophylaxis and management first as one of the most common side effects (more than hypersensitivity)
We moved “The cholinergic syndrome induced by govitecan, can be prevented by administering atropine before infusion. Optimizing patient education on managing diarrhea is necessary before starting treatment and should be reiterated if symptoms appear “ to line 233-235, to start the AE’s management with diarrhea.
- For neutropenia, the last sentence prior to management of AE section should likely move down
We moved “Primary prophylaxis for neutropenia with G-CSF is not routinely recommended.” to line 250-251
*Discussion / Conclusions
- It may be helpful to include that while toxicity and SDM are key, it is also important to understand context and patient population. The SG trials included only TNBC patients, while the Destiny Breast trials had both HR+ and HER2- patients. That does impact decision making for clinicians, especially with the ASCENT 04 recent data of SG + pembrolizumab
We added this comments in the discussion section of our study line 275-277: “While toxicity profiles and SDM are critical considerations, it is also essential to recognize the heterogeneity of patient populations across the pivotal trials analyzed in our study.”
- Patient reported outcomes should be discussed here, as that data is available for the Destiny Breast and ASCENT trials
We add line 273-275 “Even with 99% and 98% of any grade AE, both T-DXd and SG respectively showed a decreased risk of definitive deterioration compared to chemotherapy”
- Consider noting that these ADCs are moving up in the treatment for all subtypes of breast cancer, including the curative setting. This makes it even more important to understand adverse events of special interest (like ILD) when using drugs early in the disease course
We add lines 293-298: “As ADCs are increasingly integrated into the treatment landscape across all breast cancer subtypes, including in early-stage and potentially curative settings, it becomes essential to thoroughly characterize AE of special interest—such as PID—when these agents are administered earlier in the disease course. Anticipating and managing such toxicities in patients with fewer comorbidities and potentially longer life expectancy presents additional challenges for oncologists.”
General comment: The authors are looking to implement novel AE visualization tools to facilitate SDM, and this message comes through in the final part of the paper. It would be helpful to include that message in the introduction (including prior work using this methodology), and adding some more information about this methodology in the materials and methods section
We add in the introduction lines 122-123: “Moreover, decision aids incorporating visual elements are being developed in context of SDM, especially
for underserved patient populations”
We clarified the methodology paragraph lines 141-143: “We used a descriptive pooled analysis of adverse event frequencies,aggregating raw event counts and patient numbers across studies without formal hypothesis testing.”
We add in the methodology paragraph lines 147-148: “Radar plots visualization was generated in Python using the matplotlib and numpy packages”
Reviewer 2 Report
Comments and Suggestions for Authors
This manuscript is novel and elegantly written. It closes a gap in the lack of information in the comparison of detailed toxicities of trastuzumab deruxtecan (T-DXd) and sacituzumab govitecan (SG) in HER2-negative metastatic breast cancer (mBC) for patient-centered care. The study synthesizes the toxicities of T-DXd and SG in a detailed and yet non-overwhelming manner. This study particularly the figures would be a powerful tool to guide metastatic breast cancer patients and their physicians to choose a personalized and tailored treatment.
Minor comments:
- Reference 1 seems to be misplaced in the paper. The referenced paper “Wolff, A.C.; Hammond, M.E.H.; Allison, K.H.; Harvey, B.E.; Mangu, P.B.; Bartlett, J.M.S.; Bilous, M.; Ellis, I.O.; Fitzgibbons, P.; Hanna, W.; et al. Human Epidermal Growth Factor Receptor 2 Testing in Breast Cancer: American Society of Clinical Oncology/College of American Pathologists Clinical Practice Guideline Focused Update. JCO 2018, 36, 2105–2122, 289 doi:10.1200/JCO.2018.77.8738.” is not appropriate for the epidemiology (mortality rates) for breast cancer worldwide. This reference should be revised.
- In table 1 and 2, it’s unclear to readers (especially for a patient who would be reading this article) what the number after the comma represents. I suggest that the authors define the number after the comma in the header of each column or in the title of the table or as a footnote of the table.
- On lines 202-203, in the sentence” …. it is recommended to reduced dose of SG from 25% and use G-CSF in secondary prophylaxis”, the use of “from 25%” leads to confusion. I suggest that the authors revise the sentence and choose an appropriate preposition such as by or another proper preposition of their choice.
- On lines 204-205, the following sentence “G-CSF can be prescribed for 2 or 3 days à D4-D6 if neutropenia occurred at D1, or for 1 day in pegyled form at D9-D10 if neutropenia occurred at D8” is unclear. This sentence requires revision for clarity.
Author Response
Minor comments:
- Reference 1 seems to be misplaced in the paper. The referenced paper “Wolff, A.C.; Hammond, M.E.H.; Allison, K.H.; Harvey, B.E.; Mangu, P.B.; Bartlett, J.M.S.; Bilous, M.; Ellis, I.O.; Fitzgibbons, P.; Hanna, W.; et al. Human Epidermal Growth Factor Receptor 2 Testing in Breast Cancer: American Society of Clinical Oncology/College of American Pathologists Clinical Practice Guideline Focused Update. JCO 2018, 36, 2105–2122, 289 doi:10.1200/JCO.2018.77.8738.” is not appropriate for the epidemiology (mortality rates) for breast cancer worldwide. This reference should be revised.
Thank you for pointing this out. We changed it for the good reference.
- In table 1 and 2, it’s unclear to readers (especially for a patient who would be reading this article) what the number after the comma represents. I suggest that the authors define the number after the comma in the header of each column or in the title of the table or as a footnote of the table.
We add in the tables’ 1 and 2 titles: “in proportion (%)” and a footnote “Calculated wMean and % described in each study “
- On lines 202-203, in the sentence” …. it is recommended to reduced dose of SG from 25% and use G-CSF in secondary prophylaxis”, the use of “from 25%” leads to confusion. I suggest that the authors revise the sentence and choose an appropriate preposition such as by or another proper preposition of their choice.
We clarified it lines 243-246 “In the event of severe neutropenia – defined as grade 3 febrile neutropenia, grade ≥4 neutropenia lasting less than one wee, or grade ≥3 neutropenia resulting in treatment delay – it is recommended to reduce the dose of SG by 25% and to initiate secondary prophylaxis with G-CSF”
- On lines 204-205, the following sentence “G-CSF can be prescribed for 2 or 3 days à D4-D6 if neutropenia occurred at D1, or for 1 day in pegyled form at D9-D10 if neutropenia occurred at D8” is unclear. This sentence requires revision for clarity.
We clarified it lines 246-248: “G-CSF can be administered for 2 to 3 consecutive days between Day4 and Day 6 if neutropenia occurs on Day 1 of the treatment cycle, or as a single dose in pegylated form on Day 9 or Day 10 if neutropenia occurs on Day 8”
